# ILCs—Crucial Players in Enteric Infectious Diseases

**DOI:** 10.3390/ijms232214200

**Published:** 2022-11-17

**Authors:** Tamara Leupold, Stefan Wirtz

**Affiliations:** 1Medizinische Klinik 1, Universitätsklinikum Erlangen, Friedrich-Alexander-Universität Erlangen-Nürnberg, 91052 Erlangen, Germany; 2Medical Immunology Campus Erlangen, Friedrich-Alexander-Universität Erlangen-Nürnberg, 91052 Erlangen, Germany

**Keywords:** innate lymphoid cells (ILCs), mucosal immunity, enteric infections, cytokine responses, crosstalk between ILCs and other immune cells, tissue homeostasis and ILCs, ILC plasticity

## Abstract

Research of the last decade has remarkably increased our understanding of innate lymphoid cells (ILCs). ILCs, in analogy to T helper (Th) cells and their cytokine and transcription factor profile, are categorized into three distinct populations: ILC1s express the transcription factor T-bet and secrete IFNγ, ILC2s depend on the expression of GATA-3 and release IL-5 and IL-13, and ILC3s express RORγt and secrete IL-17 and IL-22. Noteworthy, ILCs maintain a level of plasticity, depending on exposed cytokines and environmental stimuli. Furthermore, ILCs are tissue resident cells primarily localized at common entry points for pathogens such as the gut-associated lymphoid tissue (GALT). They have the unique capacity to initiate rapid responses against pathogens, provoked by changes of the cytokine profile of the respective tissue. Moreover, they regulate tissue inflammation and homeostasis. In case of intracellular pathogens entering the mucosal tissue, ILC1s respond by secreting cytokines (e.g., IFNγ) to limit the pathogen spread. Upon infection with helminths, intestinal epithelial cells produce alarmins (e.g., IL-25) and activate ILC2s to secrete IL-13, which induces differentiation of intestinal stem cells into tuft and goblet cells, important for parasite expulsion. Additionally, during bacterial infection ILC3-derived IL-22 is required for bacterial clearance by regulating antimicrobial gene expression in epithelial cells. Thus, ILCs can limit infectious diseases via secretion of inflammatory mediators and interaction with other cell types. In this review, we will address the role of ILCs during enteric infectious diseases.

## 1. Introduction

Reflecting the constant exposure to food antigens, commensal and pathogenic microbes and their immunomodulatory metabolites, the gastrointestinal tract with its large surface represents the largest compartment of the body’s immune system. This intestinal immune system harbors various immune cells, including, amongst others, T- and B- cells, and innate lymphoid cells (ILCs) [1]. There is now also increasing evidence that the insides of the intestine, such as the intestinal microbiota, the intestinal barrier (consisting of intestinal epithelial cells (IECs)), and the residing immune cells influence physiological and pathological processes throughout the whole human body. A common chronic illness of the intestine is inflammatory bowel disease (IBD), which encompass the two main entities Crohn’s disease (CD) and ulcerative colitis (UC) [2,3]. These increasingly common disorders lead to severe intestinal symptoms such as anemia, chronic diarrhea, and rectal bleeding [4,5,6]. The multifactorial pathogenesis is influenced by genetic, immunological and environmental components [7,8], but importantly also by pathogenic or pathobiontic microbes [9]. 

Many human infectious diseases are caused by intracellular and extracellular pathogens, including bacteria, fungi, parasites and viruses. Mostly, infections can be controlled by the host immune system resulting in clearance of the pathogen after a short period of time. Particularly in immunocompromised individuals, infections with some pathogens are long lasting and can be accompanied by massive chronic inflammation and even morbidity and mortality during disease [10]. Mostly, pathogen infection starts at barrier and mucosal surfaces such as the intestinal mucosa, the skin and the lungs. Typically, pathogen invasion results in acute inflammation that is marked by rapid secretion of alarmin-like molecules, which orchestrate a cascade of inflammatory events including the recruitment and activation of innate lymphoid cells (ILCs). ILCs are due to their preferential anatomic location at barriers (e.g., skin, lung and intestinal mucosa) and their receptor repertoire able to quickly react to microbial and inflammatory changes with the secretion of cytokines, thereby limiting pathogen spread and excessive tissue injury [1]. They respond to numerous environmental stimuli and contribute to a swift orchestration of an innate immune response against pathogens such as helminths, extracellular bacteria, viruses and apicomplexans. Moreover, ILCs are able to interact with non-immune cells such as epithelial cells and stromal cells, and may thereby induce tissue reorganization [11]. However, some studies also suggest an involvement of dysregulated ILC responses in the pathogenesis of chronic inflammatory disorders and tissue destruction [1]. Based on their master transcription factors and cytokine expression profiles, ILCs have been divided into major groups and respond similar but earlier than their adaptive T cell counterparts Th1, Th2, Th17 and cytotoxic T cells [12]. Importantly, many studies also demonstrated that ILCs are bridging the innate and adaptive immune system. Innate cells in the myeloid compartment directly sense invading pathogens and produce inflammatory cytokines to activate ILCs, which can directly answer by secreting effector cytokines to orchestrate local immune responses [13]. Moreover, they have the ability to present antigens to Th cells through MHCII, therefore promoting Th cell differentiation and effector function [13,14,15,16,17]. 

In this review, we will highlight the interplay of ILCs with other cells during infectious diseases in the gut, caused by invading intra- and extracellular pathogens, such as apicomplexa, helminths, bacteria and viruses. Furthermore, we will address the role of ILCs in intestinal homeostasis and inflammation and provide a short outline about the role of ILC plasticity in these processes. 

## 2. The Different ILC Subsets

Based on their transcription factor profiles and cytokine secretion patterns, efforts to categorize both human and mouse ILCs broadly divided them into cytotoxic (NK cells) or non-cytotoxic ‘helper’ ILCs (ILC1, ILC2, ILC3) [18,19]. NK cells and ILC1s have several characteristics in common, including their expression of the transcription factor T-bet, the natural cytotoxicity receptor (NCR) NKp46 and the ability to produce large amounts of interferon-gamma (IFNγ). While NK cells play important roles in the context of immunosurveillance and are rather circulating cells, ILC1s display no or only weak cytotoxic activity, are largely tissue-resident and their development does not depend on eomesdermin (EOMES) [20]. Substantial numbers of NK cells and ILC1s are present in the intestine and the liver and have been implicated in host protection against infections with viruses, bacteria and protozoa. Unlike NK cells, ILC1s do not express class I MHC-specific inhibitory receptors such as CD94/NKG2A, Ly49, and KIR or perforin and granzyme B. Dependent on the tissue context and their state of activation, ILC1s may express the markers CD49a, TRAIL and CD103 [20,21,22]. 

ILC2s are broadly characterized by their capacity to secrete the type 2 cytokines IL-4, IL-5 and IL-13 [15,23,24] and can be in the steady state primarily found in, e.g., skin, lung tissue, the intestinal lamina propria and fat-associated lymphoid clusters of the intestinal mesentery [16]. They swiftly respond to the alarmin-like cytokines IL-25, TSLP and IL-33 and are strongly involved in the innate immune response to metazoan parasites, such as hookworms. After resolution of infection, ILC2s also help to repair the damaged tissues via the production of amphiregulin (AREG) [25,26]. Furthermore, these cells are also involved in the pathogenesis of asthma and allergy by secretion of type 2 cytokines [27,28,29]. ILC2s express high levels of GATA-3, and lack of GATA-3 prevents the development and function of this ILC subtype [30,31]. Of note, the development of ILC2s is also depending on the transcription factor retinoic acid receptor related orphan receptor (RORα) [32]. Until now, only limited data is available on the role of ILC2 in inflammatory diseases of the gastrointestinal tract. Hitherto, in murine DSS colitis models, ILC2s activated by IL-33 were demonstrated to have a protective effect [33,34]. 

ILC3s are characterized by their expression of the transcription factor RORγt and depend on IL-7 for their development and function [35,36]. They are abundantly located in the intestinal mucosal tissue where they are important for intestinal homeostasis, inflammatory responses and clearance of extracellular pathogens [37]. ILC3s are the most heterogeneous group of ILCs and can be subdivided into three subtypes, depending on their function [38,39]. Lymphoid tissue inducer (LTi) cells are embryonically emerging and are required for the organogenesis of lymph nodes and peyers patches [38,39]. In mice, LTi cells express CD117, CCR6, CD4 and CD127. Human LTi cells are similar to murine LTi cells but do not express CD4 [40,41]. In response to TNF-α and lymphotoxin-β stimulation during embryogenesis they assist in building lymphoid organs. LTi cells can also secrete IL-17A and IL-22 to help defend the intestine against invading pathogens. In addition to LTi cells, ILC3s can be subdivided into NCR^+^ ILC3 and NCR^−^ ILC3 with respect to their expression of the natural cytotoxicity receptors NKp46, NKp44 and NKp30 [37,41]. ILC3 can also express the CC-chemokine receptor 6 (CCR6; also known as CD196) and this marker is used for further categorization into CCR6+ LTi-like cells and CCR6- ILC3 lineages [42,43]. NCR+ ILC3s in mice express CD117, CD127 and NKp46, whereas human ILC3s additionally express NKp44. Besides, mouse NCR- ILC3s differ from human NCR- ILC3s in their lack of CD117 [41,44]. 

## 3. ILC Plasticity

Similarly, to what is known for Th cell subsets, also the phenotype of ILC subsets is not rigid and can be changed upon alterations in their microenvironment. This phenomenon is called plasticity and describes the function of developing cells and mature cells to alter their functions and phenotype in response to changing physiological and pathophysiological stimuli [45,46,47]. In 2010 it was shown for the first time that human ILC3s vary their cytokine profile depending on the way they are stimulated [48,49]. In the mouse intestine it was reported that the transition of ILC3s that secrete IL-22 into IFNγ-producing ILC1s is regulated by inflammatory signals [50,51]. In the murine intestine NKp46- ILC3s express only RORγt, while NKp46+ ILC3s can express RORγt and T-bet, which is needed for their differentiation from NKp46- ILC3s [42,46,52]. During infection of mice with *Salmonella* Typhimurium and in response to IL-12, NKp46+ ILC3s can downregulate expression of RORγt and differentiate into ILC1s. In the inflamed part of the intestine in Crohn’s disease, ILC1s were shown to be increased at the cost of ILC3s, which is indicating that chronic inflammation can lead to the transdifferentiation of ILC3s into ILC1s [21,53,54,55]. This is supported by findings in acutely infected mice with a human immune system, where a swift conversion of human ILC3s into ILC1s was detected [53]. This differentiation is suggested to be caused by IL-12, which induces the downregulation of RORγt expression and leads to upregulation of T-bet expression in vitro. Additionally, it has been demonstrated that this transdifferentiation of human ILC3s into ILC1s initiated by IL-12 is reversible, since IL-1β and IL-23 can convert ILC1s back into IL-22-secreting ILC3s. This effect is enhanced by the vitamin A metabolite retinoic acid (RA) [46,53]. Overall, it is hypothesized that in the intestine and similar mucosal tissues where ILC3s are the main ILC population under steady-state conditions the driver of ILC3 transdifferentiation into ILC1 is acute inflammation [46]. After inflammation is resolved ILC1s can convert back into ILC3s [46,53].

Additionally, it was shown that transcription factors play a role during ILC plasticity, since the transdifferentiation from ILC3 to ILC1s for example is depending on the balance of RORγt and T-bet [42]. A recently published study by Krzywinska and colleagues, further showed a role for transcription factor HIF-1α regulating plasticity of NKp46+ ILCs in the gastrointestinal tract by controlling ILC phenotype and function [56]. Hypoxia-inducible transcription factors (HIFs) mediate cellular adaption to low oxygen levels. The gut mucosa, due to its close proximity to the oxygen depleted lumen, is characterized by hypoxia [56,57]. In IBD, hypoxia in the intestinal mucosa was shown to be extensively worsened. Additionally, the balance between NKp46+ ILCs (ILC3s secreting IL-22 and ILC1s secreting IFNγ) plays an important role in gut homeostasis [53,56,58]. Krzywinska et al. show by scRNA-seq that deletion of HIF-1α in NKp46+ ILCs promotes an ILC3 phenotype and that HIF-1α contributes to ILC3-to ILC1 conversion in NKp46+ ILCs. Consequently, also the expression of IL-22-inducible, gut-homeostatic genes, such as *Reg3*γ, *Defa21* and *Muc2* were enhanced in HIF-1α KO mice due to higher ratio of IL-22 expression [56]. Indeed, since ILC3-derived IL-22 is known to influence the microbiome, in this study, the HIF-1α KO mice display an increase in homeostatic Bacteroidetes species, as well as a decrease in potentially pathogenic Firmicutes species in the colon during steady state conditions. This study highlights the role for transcriptional control in ILC subsets. This HIF-1α-dependent enhanced ILC1 to ILC3 conversion and the strengthening of the ILC3/IL-22 driven prohomeostatic properties of the gut mucosa, whilst simultaneously, the ILC1 proinflammatory responses are dampened, results in a better outcome of disease in the context of DSS-induced colitis, shown by decreased histology scores and reduced weight loss in mice with HIF-1α KO in NKp46+ ILCs. Strikingly, this indicates a therapeutic role in IBD for transcriptional control of HIF-1α in ILC subsets and thereof a possible improvement in inflammatory processes [56]. 

The master transcription factor GATA-3 of ILC2s controls IFNγ expression and thereby restricts IFNγ secretion and the plasticity towards ILC1s [59]. However, IL-1β and IL-12 were shown to induce the conversion of ILC2s to ILC1s in vitro [28]. In mice infected with influenza virus, the shift of ILC2s towards ILC1s has been demonstrated to happen in the lungs. Due to viral challenge, lung ILC2s can demonstrate plasticity and adopt an ILC1 phenotype with substantially lower expression of GATA-3 but production of IFNγ, whereby IL-12 and IL-18 modulate this conversion [28]. In the same study, it was shown that ILCs cluster with influenza virus-positive epithelial cells within the inflamed region and acquire the phenotype of ILC1s. Interestingly, Silver and colleagues reveal that myeloid cells express IL-12 and IL-18 mRNA in close proximity to ILCs in the inflamed area, highlighting the communication of myeloid cells with ILCs, which potentially drives phenotypic conversion of ILC2s to ILC1s, which in turn can exaggerate local inflammation [28]. 

Collectively, this plasticity enables ILCs to adapt to changing local conditions, which is necessary to adjust responses to different pathogenic insults. Since ILCs are mainly tissue resident under homeostatic conditions, their plasticity enables a swift response to alterations in the microenvironment, induced by pathogens, without necessity for new recruitment of ILC subsets to the site of infection. However, ILC subsets that, after resolution of inflammation, do not re-differentiate into their normal ILC subset may contribute to chronic pathology. Therefore, a fine-tuned balance of this ILC plasticity is required for gut homeostasis [46]. 

## 4. ILC Subsests in Intestinal Homeostasis and Inflammation

ILCs are primarily located at border surfaces such as the intestine. These border surfaces are in close proximity to the outside world and are consistently confronted with bacteria, viruses and parasites, but also with commensal microbes, food antigens and metabolites. ILCs were shown to be key players in mediating immunity against such pathogens and are regulating tissue homeostasis. ILCs are distributed diversely across the human gastrointestinal tract (GI) with an increase in total ILC from the oral part to the distal intestine, with the highest numbers in ileum and colon. In humans, ILC1s are fortified in the esophagus, whereas ILC2 are only present in limited numbers in the human intestine and ILC3s increase towards the colon [60]. Mouse models showed that ILCs are crucial players in the maintenance of intestinal homeostasis with their contribution to protective immunity towards pathogens, amongst others by secreting cytokines [42,61]. *Helicobacter typhlonius*, a commensal in the murine intestine, similar to a frequent colonizer of the human stomach (*Helicobacter pylori*), is associated with gastritis and gastric cancer [36]. ILC1s are highly represented in the upper GI tract and it was shown that TRUC mice (*Tbx21^−/−^ Rag2^−/−^*) lacking T-bet expression in the innate immune compartment develop *H. typhlonius*-induced colitis, which was triggered by colonic CD103- CD11b+ dendritic cell (DC)-derived TNF-α. Moreover, this TNF-α is synergizing with IL-23, which activates innate immunity and drives IL-17 secretion by ILCs [36,61]. However, upon *Salmonella* infection, T-bet+ ILCs are the major producers of IFNγ, which triggers the secretion of mucus-forming gylcoproteins necessary for protection of the epithelial barrier [36,42]. Additionally, ILC1s are the key producers of IFNγ during *T. gondii* infection and it was shown that T-bet-deficient mice fail to control parasite growth [62]. 

ILC2s are known to promote regeneration of injured gut-tissue during inflammation. In acute DSS colitis, it was shown in mice that IL-33 stimulation triggers ILC2s to secrete AREG, a ligand of the EGF receptor. Activated EGF signaling led to an enhanced representation of mucus-forming goblet cells and goblet cell-mediated mucus-production, which is crucial for protection of the intestinal epithelium against inflammation [25]. 

Recently, it has been shown that signals from the nervous system contribute to the activation of ILC2s. The neuropeptide neuromedin U (NMU) is produced by cholinergic enteric neurons and can stimulate or boost the expression of IL-5 and IL-13 by ILC2s, as they express Neuromedin U receptor (NMUR1) themselves. This was shown to simultaneously increase resistance to helminth infection in the intestine [63,64]. Similarly, vasoactive intestinal peptide (VIP) produced by the enteric nervous system (ENS), influences the expression of IL-5 by gut and lung ILC2s [65]. VIP is a member of the glucagon-secretin superfamily of peptides and is produced in the intestine after feeding [66]. When binding to its receptors VIPR1 and VIPR2, VIP assists in digestion in the intestine [67]. In the study of Pascal et al., the authors show that VIP is induced via feeding and potentiates ILC2 and ILC3 effector function; both cell types highly express VIPR2 [68]. VIP alone only showed a modest effect on ILCs, however, when VIP is induced via feeding and IL-33 (or IL-25) and IL-23 (or IL-1β) are present, this induced the expression of effector cytokines IL-5 and IL-22 by ILC2s and ILC3s, which in turn also refers protective immunity towards type 2 (helminths) and type 3 (*C. rodentium*) associated pathogens. This communication between the ENS and ILCs and subsequent potentiation of ILC effector function due to potential dangers that could come with food intake, helps maintaining intestinal homeostasis. In the same study, they recognized that enteroendocrine cells secrete cholecystokinin (CCK) by sensing the presence of food (and the danger for potential foodborne pathogens), which boosts the release of VIP by the ENS within minutes [68]. This fast communication between epithelial cells, the ENS and ILCs is a crucial network for retaining homeostasis and a defense mechanism that is waiting in the wings to react to potential danger signals due to food intake.

ILC3s, which are mainly free from pattern recognition receptors [69], are activated after infection by cytokines, such as IL-1β and IL-23, secreted from epithelial and in particular myeloid cells [11] and, subsequently, release IL-17 and IL-22 to protect the host [70,71,72,73]. The cytokine IL-22 has been shown to act mainly on non-hematopoietic cells such as epithelial cells, keratinocytes, and stromal cells. ILC3s are key producers of IL-22 and the primary source of this cytokine at the steady state [74,75]. Furthermore, IL-22 is a crucial player for homeostatic balance in the intestinal mucosa [76]. RegIII proteins belong to the group of antimicrobial lectins that are needed for protection against bacterial infections and their expression is triggered by IL-22 [77,78]. Many studies attribute ILC3-derived IL-22 protective immunity against various bacteria by directly regulating antimicrobial gene expression in epithelial cells [73,76,79]. Further, studies showed that production of IL-22 is depending on signals from the microbiota [80].

Tumor necrosis factor (TNF) is a cytokine, which regulates host defense and tissue regeneration under homeostatic conditions and is secreted by many cell types. However, if TNF is overexpressed in the intestine, it targets the intestinal epithelium and leads to increased cell death, following tissue inflammation [81,82]. In the context of IBD, blocking TNF can lead to a therapeutic benefit [83]. Zhou et al. determined a new feature of ILC3s, which is tissue protection from TNF-induced inflammation via the secretion of heparin-binding epidermal growth factor-like growth factor (HB-EGF) [82]. In this study, ILC3s notice tissue inflammation or damage of the intestinal epithelium via released IL-1β, which boosts ILC3 response, following HB-EGF secretion and thereof protection from TNF-induced epithelial cell death in the intestine. This feature of HB-EGF secretion of ILC3s complements studies demonstrating that ILC2s have the property of secreting AREG, which can promote the repair of intestinal and lung epithelium [82]. Strikingly, HB-EGF production in ILC3s is independent from IL-22, since transient blockade of endogenous IL-22 does not affect protection from TNF-induced cell death via HB-EGF. Importantly, ILC3-derived HB-EGF can limit acute and chronic intestinal inflammation and the authors report a reduced HB-EGF + ILC3 population also in the inflamed intestine of IBD patients, leading to the suggestion that loss of ILC3s can increase susceptibility of the intestine towards TNF-induced epithelial damage [82]. This study sheds a light on a, so far, unknown protective feature of ILC3s towards excessive TNF production in the intestine, which can be restricted by HB-EGF [82]. With these two branches of protection—on the one hand via secretion of IL-22 due to bacterial evasion, and on the other hand via production of HB-EGF necessary to protect against TNF-induced cell death- ILC3s arise as innate key players in maintaining homeostasis by supporting/orchestrating barrier repair mechanisms of the intestinal epithelium. However, one has to bear in mind that a tight regulation of the secreted amounts of these cytokines is important, since excessive IL-22 for example, can lead to tumor growth and inhibition of apoptosis in the microenvironment of colon cancer and ulcerative colitis [84].

To sum up, ILCs are crucial players in homeostasis and inflammation by secretion of inflammatory mediators and influence on other immune and non-immune cells. Latest studies also highlight the importance of ILCs for host defense against various pathogens [1], which will be part of the following section. 

## 5. ILCs in Immune Response to Gut Intracellular Pathogens

Recent studies underline the importance of ILCs for host defense against some primarily intracellular pathogens [85,86]. In this section, we will discuss the role of ILC1s during *T. gondii* and *Salmonella* infection. 

### 5.1. *Toxoplasma gondii*

*Toxoplasma gondii* (*T. gondii*) is an obligate protozoan parasite belonging to the phylum Apicomplexa, which can infect warm-blooded vertebrates, resulting in the disease called “toxoplasmosis” [85,87]. Ingestion of contaminated food with for example *T. gondii* tachyzoites leads to acute infection [88,89]. Furthermore, developmental stages of *T. gondii* can disseminate and establish latent infections in brain and muscle tissue [90]. When infection occurs orally, the intestine is the first port where *T. gondii* meets the host immune system. In the intestine, *T. gondii* infects the intestinal epithelium [89,91].

After infection with *T. gondii*, DCs, macrophages and neutrophils are activated and secrete proinflammatory cytokines, including IL-12, which stimulates NK cells, ILCs and T cells to secrete IFNγ [92,93]. It was shown in mice, that the cytokines IFNγ and IL-12 are crucial for protection against this parasite [94,95,96]. NK cells are suggested to be important for host resistance at early onset of *T. gondii* infection, while adaptive immunity contributes to protection during the chronic phase of infection [97]. Furthermore, NK cell-derived IFNγ enhances differentiation of monocytes into inflammatory IL-12-producing DCs, necessary for protection [98] (reviewed in detail in [99]).

Only recently, also ILC1 have been shown to support clearance of *T. gondii* by the host [62,100]. Klose et al. demonstrated that mice deficient for T-bet secrete less IFNγ due to an ILC1 deficiency, whereas NK cell numbers remained the same in the small intestine during *T. gondii* infection, suggesting that ILC1 are a main source of IFNγ in the small intestine during infection, apart from NK cells or NKp46+ NK1.1+ ILC3s [99,100]. Recent studies emphasize the impact of ILC1s and NK cells in recruitment of inflammatory monocytes and DCs to protect the host against *T. gondii* [98,100]. Additionally, it was demonstrated by Clark et al. that infection with this apicomplexan parasite stimulates stromal cells to produce the alarmin IL-33, and together with IL-12, this amplifies IFNγ secretion by ILCs, necessary for protection [101]. All in all, these studies indicate a crosstalk of ILC1s, stromal cells and DCs, conveyed by the inflammatory mediators IL-12, IL-33 and IFNγ, which is crucial for host protection against *T. gondii* [99].

Mainly, ILC1 and NK cells often work closely together to ensure innate host protection, however, recently, a conversion of NK cells into ILC1-like cells during *T. gondii* infection was described for the first time [102]. The “ex NK cell”-ILC1-like cells were described to be distinct from steady state NK cells and ILC1s in uninfected mice. Moreover, *T. gondii* induced ILC1-like cells were not tissue resident, like it was demonstrated before for tumor induced NK cell conversion into ILC1-like cells, which are limited to the tumor microenvironment. In contrast, transdifferentiated ILC1s were able to circulate during inflammatory conditions [102,103]. Strikingly, after infection was resolved, the ILC1-like cells were maintained, like it is known for immune memory NK cells [99]. This study shed light on a, so far, underrated plasticity between NK cells and ILC1s. However, the mechanism behind this transdifferentiation remains unknown and needs to be further elucidated [99,102]. 

In the intestine, also ILC3s are suggested to play a role during *T. gondii* infection, since they can limit T cell activation and immune pathology dependent on aryl hydrocarbon receptor signaling [89,104]. Recently, it was found that MyD88 signaling has an impact on ILC1s since the frequency of IFNγ + ILC1s in the absence of MyD88 significantly decreased, whereas IFNγ secretion of ILC3s was not MyD88-dependent [89]. In a subsequent study, the working group further examined the influence of the intestinal microbiota on ILC1 and ILC3 in *MyD88^−/−^* and *MyD88^+/+^* mice after oral infection with *T. gondii* [105]. Snyder and colleagues found that in the lamina propria, IFNγ expression of ILC1s was decreased in infected MyD88 KO mice as well as after microbiota depletion. Furthermore, they also recognized a double positive ILC population (RORγt+ T-bet+), which was found to be controlled by MyD88 signaling, as well as by the intestinal microbiota. This highlights the effect of the microbiota on the sustenance of ILC subsets, however one has to bear in mind that antibiotic administration has an impact on the overall physiology of the intestine which in turn also impacts ILC populations [105]. When looking at ILC populations in the peritoneal cavity after i.p. administration of *T. gondii*, interestingly, Snyder et al. found that ILC1s showed a noticeable increase in the proliferation marker Ki67 after infection, which indicates that ILC1s rather proliferate in situ, than are recruited from other parts at mucosal tissues of the body [105]. Until now, the requirement of IFNγ-secreting ILC1 and NK cells necessary for host immunity was mainly shown in murine models of *T. gondii* infection. However, the role of ILC in human *T. gondii* infection still remains to be clarified [99].

### 5.2. Salmonella Typhimurium

Subgroups of the bacterial species *Salmonella enterica* can cause “salmonellosis”, a food-borne infectious gastroenteritis in humans. The disease is usually self-limitingwhen caused by *Salmonella* Typhimurium and *Salmonella* Enteritidis. However, disease severity is dependent on the immunological status of the individual and particularly in immunocompromised individuals the course of disease during *Salmonella* infection can be fatal [106]. Common entry points for the bacterium are microfold (M) cells and DCs in PPs, but also other cells of the intestinal epithelium [99]. When entered, the bacteria can disseminate and replicate in the spleen, liver and phagocytic cells in bone marrow [107,108]. Studies demonstrated that in the small intestine, innate IFNγ is a crucial factor to control bacterial loads and systemic spread [109]. The main innate source of IFNγ are NKp46+ T-bet+ ILCs [42]. Via cell-fate mapping studies it was revealed that *Salmonella* infection may induce transdifferentiation of NKp46-ILC3s to IFNγ-producing ILC1s and that T-bet is necessary for conversion of NKp46+ RORγt+ ILCs from NKp46-ILC3s, since T-bet deficient mice (*Tbx21^−/−^*) lack NKp46+ T-bet+ RORγt+ ILCs and have reduced numbers of IFNγ-producing cells [42,99]. Moreover, it was shown that the cytokine IL-12 is playing an important role for adequate IFNγ production during infection. Although IL-12*^−^*^/*−*^ mice exhibit normal numbers of T-bet+ RORγt+ ILCs, the secretion of IFNγ is severely hampered suggesting that IFNγ production by NKp46+ RORγt+ ILC3s is dependent on IL-12 and T-bet expression [42,99]. 

Nkp46+ RORγt+ ILC-derived IFNγ is required for bacterial clearance, however it can also cause epithelial damage, since *Tbx21^−/−^* and *IFNγR1^−/−^* mice exhibit a decrease in intestinal pathology 48h post infection with *Salmonella* [42,99]. Therefore, IFNγ-secreting ILCs seemingly act as double-edged sword in pathology during *Salmonella* infection, having both favorable and unfavorable effects. 

Nevertheless, for protection against *Salmonella*, NCR- ILC3s have been shown to be crucial factors, due to induction of RegIIIβ and RegIIIγ antibacterial proteins by epithelial cells, mainly via IL-22 secretion [110]. It is known that ILC mainly reside in the tissue, however, a recent study by Kastele et al., showed that a minor ILC population can pass from the intestine to the MLN in steady state, as well as in inflammatory state. In *Salmonella*-infected mice, increased numbers of migratory RORγt+ T-bet+ ILC1s were present in the lymph nodes, which served as early source of IFNγ [99,111]. Furthermore, *Tbx21^−/−^* mice that lack IFNγ-secreting Nkp46+ RORγt+ ILCs were shown to have decreased mucus production and additionally, mice that were infected with *Salmonella* showed fewer goblet cells filled with mucus [42]. In line with this, a study from Zarepour and colleagues demonstrated that MUC2-deficient mice exhibit an enhanced susceptibility to *Salmonella* infection. Further, studies with depletion of ILCs indicate a reduced IFNγ and mucus secretion after *Salmonella* infection [112]. These studies suggest that goblet cells are regulated by IFNγ-secreting ILCs during *Salmonella* infection, but still, the detailed mechanisms remain to be elucidated [99].

Interestingly, it was shown recently, that S. Typhimurium can exploit ILC3-derived IL-22 for its own benefit to promote the infection [113]. S. Typhimurium was able to selectively enhance IL-22 secretion by ILC3s but not by T cells. This enhanced ILC3-mediated IL-22 production promotes Salmonella infection in mice. Additionally, S. Typhimurium was able to invade ILC3s, which causes caspase1-mediated ILC3 pyroptosis [113]. In turn, it was demonstrated that caspase-1 deletion in ILC3s leads to increased ILC3 survival, increased IL-22 production and thereof enhanced S. Typhimurium infection. Thus, induction of ILC3 death represents a newly discovered host defense mechanism as response towards S. Typhimurium exploiting ILC3s and ILC3-derived IL-22 [113].

## 6. ILC2s and Helminths

Helminthic infections represent a serious public health concern affecting one third of the global population particularly in developing countries [114,115]. These high numbers of infections are driven by a broad diversity of helminths that are responsible for human disease including, but not limited to, *Necator americanus* (hookworms), *Ascaris lumbricoides* (roundworms), *Trichuris trichiura*, *Trichinella spiralis* (whipworms) and different *Schistosoma* species. Helminthic infections often lead to chronic infections resulting in anemia, malnutrition, growth impairment, and immunopathology [116]. Mainly, helminths are ingested from both, humans and animals via eggs or larvae present in contaminated food or water [115]. Helminth infection can cause profound tissue damage as they migrate through different organs, such as lung, intestine, liver and skin, thereby completing their life cycle. The helminth-caused wounds and their secreted products results in activation of distinct hematopoietic and non-hematopoietic cells, leading to the initiation of type 2 immune responses [117]. Research of the last decade highlighted the role of ILC2s as they seem to be one of the most important immune cell populations in combating helminth infections [118]. Mainly, the conducted studies rely on different animal models of infection including *Nippostrongylus brasiliensis*, *Heligmosomoides polygyrus* (mimicking *Ascaris lumbricoides* and *Necator americanus* infection), *Trichuris muris* (resembling *Trichuris trichiura* infection), *Schistosoma mansoni* (mimicking infection with *Schistosoma* spp.) and many more [119]. The following section will give an overview of the crucial role of ILC2s in helminth infections.

### 6.1. Nippostrongylus brasiliensis

The majority of studies addressing the role of ILCs during helminth infection utilize the *Nippostrongylus brasiliensis* (NB) infection model, which became a standard infection model for determining type 2 immunity due to its simple life cycle and provocation of strong type 2 immune responses [118]. During the last years, it has become evident that type 2 immune responses evolved to limit parasite burden, as well as to suppress overwhelming inflammation and to mediate a swift repair of injured tissues [120]. Infection with NB primarily happens via skin penetration and in the case of laboratory adaption, via subcutaneous injection of infective third-stage larvae (L3). Subsequently, the parasite enters the blood vessel, and thereby is carried to the lungs where it penetrates the lung parenchyma leading to hemorrhagic inflammation. The parasite molts in the lung and the emerging L4 larvae migrate via the trachea to the intestine. The last molt occurs in the intestine resulting in the L5 larvae (mature adult), which are found lightly attached to the proximal half of the small intestine. In the following, eggs are produced earliest at 6 days post infection [118]. Immunocompetent mice expel the parasite at 9–12 days post inoculation via the so-called weep-and-sweep mechanism, which is characterized by IL-13-induced epithelial cell turn over, goblet cell hyperplasia and mucus secretion, and increased muscle-contractility [121].

Following tissue damage due to hookworm penetration, profound amounts of alarmins such as IL-25, TSLP and IL-33 are secreted, triggering the activation of ILC2s [122,123] (Figure 1). During NB infection, intestinal tuft cells were shown to be the main source of IL-25, which promotes IL-13 secretion by ILC2s (Figure 1). Thereafter, ILC2-derived IL-13 induces tuft cell hyperplasia and parasite clearance. Further, mice deficient in tuft cells suffer from delayed parasite expulsion and impaired mucus production [122,123,124]. This demonstrates that tuft cells as specialized epithelial cells coordinate anti-helminth protection via their production of alarmins, thereby activating IL-13 production by ILC2s, which finally assists in parasite clearance [125]. Additionally, intestinal tuft cells can also secrete TSLP after parasitic infection, revealed by single-cell RNA sequencing [126]. Only this year, Varyani and colleagues added another important piece to the puzzle of the ILC2-tuft cell circuit. They demonstrate that macrophage migration inhibitory factor (MIF) is needed for expansion of intestinal tuft cells during NB infection and that MIF-deficient mice show defective innate responses [127]. The lack of MIF was compensated completely by administration of IL-25, which restored tuft cell differentiation and goblet cell expression of RELM-β, indicating that MIF is needed upstream of the ILC2-tuft cell system. Macrophages as well as ILC2s were demonstrated to express MIF receptor (CXCR4), suggesting MIF to be a co-factor on both cell types to activate response to IL-25 in helminth infection (Figure 1). Collectively, MIF was shown to be an important player in intestinal immunity during helminth infection and should be added to our understanding of how protective responses towards parasites are orchestrated [127]. Additionally, the alarmin IL-33 is awarded a notable role in anti-helminth immunity. By using a mouse enteroid-immune cell coculture system, Waddell et al. demonstrated that IL-33 indirectly induces goblet cell differentiation via stimulating IL-13 secretion by ILC2s and not via direct impact on epithelial cells. Moreover, they delineate that IL-13 is necessary for IL-33 induced goblet cell hyperplasia in vivo [128]. A study from Knipfer and colleagues shows that signaling of the chemokine CCL1 to CC-type chemokine receptor CCR8, which is expressed on ILC2s, can support the accumulation and the biological functions of ILC2s in type 2 mediated inflammatory settings such as parasitic infections with NB (Figure 1). Additionally, activated ILC2s were shown to be a source of the CCR8 ligand CCL1, indicating a supporting role for tissue specific ILC2 functions by the CCL1/CCR8 axis in an auto-/paracrine manner [129].

Peptides secreted from the enteric nervous system, such as NMU and VIP induce ILC2 effector function, which in turn improve protective immunity towards helminths [63,64,68]. In contrast, calcitonin gene-related peptide (CGRP), which is also secreted from neurons (Figure 1) negatively regulates innate type 2 immunity in gut and lung [130,131,132]. After infection with NB, ILC2s induce expression of CGRP and its receptor (CGRPR). Furthermore, CGRP was shown to act as negative regulator of ILC2 responses since type 2 cytokine production and proliferation of ILC2s was decreased in response to CGRP in vitro and in vivo, indicating that neuron-derived products can be both, activators and inhibitors of ILC2 responses [131].

Research performed until now, declares that ILC2s are the main source of IL-13, but it still needs to be further deciphered whether ILC2-derived IL-13 is necessary for worm expulsion in the context of NB infections, rather than Th2 cell derived IL-13 [133]. In this context, the study of Varela and colleagues provides important insights, since ILC2^4−13ko^ mice with defective IL-4/IL-13 secretion from ILC2s show impaired worm expulsion, as compared to ILC2^WT^ mice. However, when IL-4/IL-13 is deleted selectively in T cells, mice exhibit normal NB expulsion, indicating a dispensable role for T cell-derived IL-4/IL-13 in the context of NB infections but emphasizing the role for ILC2-derived IL4/IL-13 in the same setting [134].

### 6.2. Trichuris muris

*Trichuris muris* (TM) is used as an animal model that mirrors the features of human whipworm infection. TM is a non-migrating parasite that resides in the lumen of the large intestine [135]. Similar to other nematodes, the life cycle of TM starts with ingestion of infectious eggs released in feces of infected hosts. After intake of infectious eggs, they reach the intestine and accumulate in the caecum. Later on, the eggs develop to L1 larvae in the lower gastrointestinal tract of infected mice. The larvae molt several times until reaching L4 larvae and the adult form of TM can be detected in caecum and proximal colon of the infected animal [135].

NB and TM infection models have in common that intestinal tuft-cell derived IL-25 promotes IL-13 release by ILC2s, which in turn elicits further expansion of IL-25 producing tuft cells [124]. This tuft cell hyperplasia is dependent on chemosensory taste receptors, namely the transient receptor potential cation channel, subfamily M, member 5 (Trpm5), which leads to release of acetylcholine and activation of nearby vagal nerve fibers [125]. Similarly to the above-mentioned murine infection model NB, also IL-25 seems to play a role in parasite expulsion in TM infection [125]. TM-infected mice, treated with recombinant IL-25, exhibit enhanced parasite expulsion [124]. 

The epidermal growth factor (EGF)-like molecule, amphiregulin (AREG) has been shown to play an important role in orchestrating host resistance but also tolerance mechanisms. Earlier, AREG was characterized as epithelial cell-derived factor, but recent studies show that AREG is also expressed by several activated immune cell populations under inflammatory conditions (in detail reviewed in [136]). It was shown before, that Th2 cells express AREG, which was required for protective immunity after TM infection. In this study, AREG-deficient mice exhibit delayed expulsion of the nematode TM [137]. Similarly, also ILC2s secrete AREG and thereby promote tissue repair after helminth infection [138]. 

VIP, which is produced by the ENS, was shown to be induced by feeding [66]. Further, when the cytokines IL-33 and/or IL-25 are present, VIP potentiates the production of ILC2-derived effector cytokines, via increased amounts of cAMP, and via mobilization of energy by glycolysis. Due to this potentiation, VIP increases resistance to TM [68]. By using knock out mice with deleted VIPR2 (receptor for VIP) in ILC2s, Pascal and colleagues report a significant increase in worms in the caeca of these mice, compared to controls 14 days post infection. Additionally, the number of IL-5 expressing ILC2s were significantly declined in the KO mice. Strikingly, the production of IL-5 by ILC2s and the following recruitment of eosinophils and the general resistance to infection must have been mediated via ILC2s, since only little numbers of T cells were detected in the intestine at this time point, as reported by the authors [68]. This emphasizes the importance for communication between the ENS via neuropeptides with ILC2s. Additionally, since only little numbers of T cells were identified, this highlights the crucial role for ILC2s orchestrating immune responses against TM infections to protect the host and expel the parasite.

Collectively, these studies highlight a complex interplay between inflammatory mediators, and immune cells with ILC2s that promote a fine-tuned immune response against helminths, leading to pathogen clearance. 

## 7. ILC3 Respond to (Extracellular) Bacteria

ILC3s are mainly located in mucosal tissues and were shown to be essential in the initiation of appropriate immune responses upon pathogenic infection [139,140]. In the next section, we will focus on the murine infection model with *Citrobacter rodentium* (CR) and intestinal infections with *Helicobacter* spp.

### 7.1. Citrobacter rodentium

*Enteropathogenic* (EPEC) or enterohaemorrhagic *Escherichia coli* (EHEC) infections are amongst the main causes of persistent diarrhea in children from developing countries [141]. By forming attaching and effacing (A/E) lesions, EPEC and EHEC can adhere to the intestinal mucosa, characterized by intimate bacterial attachment to the intestinal epithelium [142] (Figure 2). Via their type 3 secretion system (T3SS), they inject bacterial effector proteins into the host cell’s cytosol, which leads to alterations in many host cell responses, resulting in disruption of homeostasis of the intestinal tract [143]. *Citrobacter rodentium* (CR), a pathogen naturally occurring in rodents, was shown to resemble the T3SS-induced pathogenesis of the intestinal mucosa and host immune responses towards EPEC/EHEC in infected humans, thereof this model is used to study host-immune response in the context of A/E pathogens ([140], (detailed reviewed in [144]). After oral inoculation of mice with CR, the bacterium colonizes the caecum and colon and proliferates rapidly [143]. CR adheres to intestinal epithelial cells and induces colitis, which causes profound intestinal dysbiosis [145]. Of note, CR-induced colitis is dependent on the genetic background of the host: in immunodeficient mice infection can be fatal, whilst in C57BL/6 mice it causes self-limiting inflammation of the caecum and colon [146,147]. 

Defense, to prevent the invasion of enteric pathogens during the early phase of infection, is established via innate immune responses. This defense includes natural host barriers such as mucus layer, antimicrobial peptides secreted by epithelial cells, and phagocytic cells [148]. Many cytokines are released in response to CR infection by numerous host cells to protect the host from invasion. Moreover, several mechanisms such as enhancement of intestinal barrier function and recruitment of neutrophils are triggered by infection [76,149,150,151]. 

One important discovery with respect of delineating mucosal defense mechanisms in response to CR infection is that IL-22 plays an essential role in maintaining gut epithelial barriers [152]. The cytokine IL-22 directly affects non-hematopoietic cells such as epithelial cells, keratinocytes and also stromal cells [75]. For ILC3s it was shown that they release IL-22 in high levels, and that they are the only source of IL-22 in the steady state gut. IL-22 release by ILC3s drives the expression of the antimicrobial lectin RegIII in epithelial cells, which is necessary for sequestering bacteria away from epithelial cells [77] (Figure 2). This suggests that ILC3s and ILC3-derived IL-22 are key players in immunity to bacteria by directly regulating antimicrobial gene expression in epithelial cells, facilitating bacterial clearance [76]. 

Only recently, it was reported by our group that the transcription factor IRF-1 controls ILC3 numbers [153]. IRF-1, which is actually known to be important for the control of intracellular pathogens via driving protective type 1 immunity, was also shown to be activated during typical type 3 immune responses directed against CR infection [153]. IRF-1 activates IL-22 production in CCR6+NKp46- and CCR6-NKp46+ ILC3s (Figure 2), thereby preventing CR dissemination in the colon. In more detail, IRF-1 induced the expression of IL-12Rβ1 chain in ILCs, thereby allowing ILC3s to respond to the cytokines IL-12 and IL-23, which are produced by myeloid cells due to CR infection [154]. These findings emphasize the importance of a functioning cross-talk of the myeloid compartment with ILC3s, to protect from CR infections, as well as the importance of transcription factors for fine-tuning these communications, which are necessary for inducing proper responses towards pathogen invasion, which is mainly noticed by the immune system via secreted alarmins (such as IL-12 and IL-23). 

ILC3s were also shown to react to bacteria-fermented dietary compounds, such as short-chain fatty acids (SCFAs), via their receptor GPR43 (Figure 2), thereby increasing IL-22 production [80]. However, when GPR43 is missing, as shown in knock-out mice, ILC3 proliferation is dampened and ILC3-derived IL-22 levels are reduced, consequently rendering mice more susceptible to gut injury and infection, indicating a microbiota-dependent regulation of ILC3 function [80].

Apart from dietary compounds or vitamins, which are important for ILC3 function and phenotype [80,155], more and more studies report a crucial role for neurotransmitters and neuropeptides (e.g., neuromedin U, acetylcholine), produced by neurons, in regulating the function and numbers of ILCs during homeostasis and inflammation, highlighting a neuro-immune communication [63,64,156]. The enteric nervous system (ENS) produces vasoactive intestinal peptide (VIP), which was shown to have immune regulatory functions, via VPAC1 and VPAC2 (two G protein coupled receptors) engagement [157]. Previously, VIP was found to be required for postnatal formation of lymphoid tissues such as cryptopatches, which are small clusters composed of ILC3s [156], which directly links the VIP-producing-ENS with ILC3s. ILC3s express homing receptors, such as CCR9, which modulate their migration to the gut. The expression of CCR9 in ILC3s can be upregulated in response to retinoic acid (RA), a metabolite of vitamin A [158,159]. In the study of Yu and colleagues, VIP was shown to regulate the local production of RA by intestinal CD103+ dendritic cells (DCs) (Figure 2), but these cells are significantly reduced in *Vip^−/−^* mice, when compared to control mice. The decrease in RA production from the significantly reduced CD103+ DC cell population leads to a reduced CCR9 expression in ILC3s and therefore less ILC3s migrate to the intestine. This loss of ILC3s renders the mice more susceptible to CR infection, as simultaneously to the reduced amount of ILC3s, less IL-22 and antimicrobial peptides, important for pathogen defense, are produced [156]. In this way, VIP, produced by the ENS, critically mediates the communication between the ENS and ILC3s. Importantly, VIP regulates ILC3 recruitment in an additional way, including a VIP-VPAC1-mediated migration of ILC3s to the intestine [156]. VPAC1 (receptor) mediates the effect of VIP on promoting the recruitment of ILC3s to the intestine. Cells that regulate the traffic of leukocytes to the intestine express VPAC1 and/or VPAC2. By generating *Vpac1^−/−^*, *Vpac2^−/−^* and *Vpac1^−/−^Vpac2^−/−^* mice the authors examine which receptor mediates the effect of VIP on enhancing ILC3 recruitment to the intestine. They found that only *Vpac1^−/−^* and *Vpac1^−/−^Vpac2^−/−^* but not *Vpac2^−/−^* mice show reduced numbers of ILC3s leading them to assume VPAC1 plays an important role in VIP mediated recruitment of ILC3s [156]. VIP not only maintains homeostasis by the sustenance of ILC3 pools in the intestine, with its role for ILC3 recruitment, but also feeding-induced VIP potentiates ILC3 effector function by synergizing with present IL-23 (or IL-1β) and increases the production of IL-22 [68]. With this potentiation of ILC3 effector function by the ENS—ILC communication, the immune system stays alert via signals from the ENS for possible foodborne pathogens, coming with food intake. These studies highlight the importance for a functioning communication of the ENS with ILCs via VIP since otherwise, when VIP is lacking, this leads to malfunctioning of defense mechanisms against bacterial pathogens. Similarly, also glial cells were shown to interact with ILC3s via glial-derived neurotrophic factor (GDNF) family ligands (GFL), which are recognized by neuroregulatory receptor RET (Figure 2), expressed by ILC3s [160]. Interestingly, RET deletion in RORγt+ ILC3s did not affect IL-17 secretion but led to impaired IL-22 production. Furthermore, mice deficient in RET showed a more severe phenotype when infected with CR [160], which further emphasizes the ILC–ENS communication when it comes to infection. 

Additionally, an interaction between fibroblastic reticular cells (FRCs) and ILCs recently was emphasized by findings of Cheng and colleagues, who describe that communication between FRCs and ILCs are required for preservation of functional ILC populations, especially ILC3s, which in turn are relevant for proper protection against CR [161]. They discovered that the absence of LTβR signaling in FRCs (by using Ccl19-Cre *Ltbr*^fl/fl^ mice) leads to a significant decrease in the absolute numbers of Lin- CD127+ ILCs. By further examining ILC proliferation and cell death markers it could be shown that the absence of LtβR signaling in FRCs affects the survival of ILCs, as expression of caspase 3/7 is significantly upregulated in these cells. Additionally, single cell transcriptomics data from FRCs and ILCs uncover important pathways needed for their interaction. LTβR-dependent ILC stimulation by FRCs requires the secretion of IL-7 to maintain ILC populations (Figure 2), since mice with FRCs deficient in IL-7 show a decrease in total ILC numbers and in all ILC subsets in the lamina propria. This underlines that the lymphotoxin-driven feed forward loop of FRC activation including IL-7 generation is crucial for the preservation of functional ILC populations, especially ILC3s, which, additionally enhance immunity during CR infections [161].

Interestingly, Serafini and colleagues describe that after CR infection, intestinal ILC3s remain for months in an activated state [162]. Upon rechallenge, these so-called “trained ILC3s” propagate and exhibit improved IL-22 responses and are able to better control the infection than naïve ILC3s, thus, contributing to long-term mucosal defense. These “trained” ILC3 may be a target in the future for enhanced treatment of disease caused by barrier-surface invading pathogens [162].

### 7.2. Helicobacter

In humans and also in murine models, many species within the genus *Helicobacter* induce pathogenic responses, particularly in immunocompromised hosts [163,164]. Gastric *Helicobacter* spp., such as *Helicobacter pylori* are considerable inducers of peptic ulcers and mucosa-associated lymphoid tissue (MALT) lymphomas [165]. There are also nongastric *Helicobacter* spp. nhabitating the intestine, which can induce strong T cell responses and promote the activation and proliferation of both effector and regulatory T cells. Nevertheless, previous studies demonstrate that ILCs participate in the immune response to *Helicobacter* spp., as well. In a study of Buonocore and colleagues, chronic infection of immunodeficient mice with *Helicobacter hepaticus* led to RORγt+ ILC3-induced gut inflammation. High levels of ILC3-derived IL-17 and IFNγ, in response to IL-23 stimulation contributed to the development of T-cell independent inflammation [70]. Likewise, ILC3s were shown to be pivotal for the development of innate-mediated colitis in an anti-CD40 mABs induced innate model of IBD [70]. Supporting the hypothesis of ILC3s being pathogenic in certain contexts of intestinal inflammation, a further study demonstrated ILC3s, being responsive to IL-23, are increased in the intestines of patients with CD. They secrete high quantities of IBD-relevant cytokines, for instance, IL-17 and IL-22, in response to IL-23 [166]. Furthermore, ILC3s were said to play a pro-inflammatory, pathogenic role in cancer [167,168]. 

Only previously, it was demonstrated in large intestines of mice lacking adaptive immunity, that introduction of the *Helicobacter* species *Helicobacter apodemus* and *Helicobacter typhlonius* led to activation of ILCs and induction of gut inflammation [165]. However, both *Helicobacter* spp. negatively regulated colonic ILC3s and induced gut dysbiosis. More strikingly, *Helicobacter* spp. treatment particularly inhibits T-bet expressing, but not T-bet negative ILC3s. Furthermore, their proliferative capacity was decreased. This study highlights a so far unknown “dichotomous regulation” of ILC3s by *Helicobacter* spp. in which this microbe can activate ILCs to produce higher amounts of cytokines (since an increase in IL-22+ and IL-17+ ILC3s was shown), leading to induction of inflammation, however, *Helicobacter* spp. treatment of mice declines ILC3 numbers in the colon, especially T-bet and RORγt expressing ILC3s. Still, further studies are needed to decipher whether the loss of ILC3s is responsible for gut inflammation or vice versa. Additionally, this *Helicobacter* infection model can help to elucidate the host-microbe interactions that substantially retain the maintenance of gut ILC3s [165]. 

Collectively, ILC3 were shown to act as double-edged sword during infectious disease. They play an important role in regulating the balance between maintenance and loss of intestinal homeostasis and contribute to immunity against intestinal pathogens. However, they can also induce inflammation which can then lead to the development of IBD or colorectal cancer. 

## 8. ILCs and Viruses 

In several studies, it was shown before that ILCs are enriched in the lung after infection with influenza virus, and that deletion of ILCs resulted in loss of airway epithelial integrity, diminished lung function and impaired airway remodeling. Additionally, several studies underpinned the relevance for ILCs in respiratory viral infections (reviewed in [169]). Respiratory infection with influenza A virus (IAV) can cause systemic inflammation and can also damage the intestine [170]. A study of Roach et al. describes an increase in intestinal tuft cells as well as in ILC1s and ILC2s during active infection with IAV. Nevertheless, when intestinal tuft cells were missing, also the increase in ILC2s was curtailed. However, the ILC2 amount in the lungs, in the absence of intestinal tuft cells, was not altered [171]. This study indicates that IAV infection of the lungs can lead to dynamic changes of tuft cells and ILCs in the small intestine and that tuft cells are required for infection-induced accumulation of intestinal ILC2s. This accumulation of tuft cells and ILC2s could be a mechanism that helps to prevent systemic infection [171]. 

Many viruses such as human immunodeficiency virus (HIV), influenza virus, rotavirus and norovirus replicate in epithelial cells to establish infections in their hosts [172,173,174]. Particularly IFNs are believed to play a crucial role in antiviral defense [175]. IFNs are released by virus-infected epithelial cells and thereby induce anti-viral state in cells that receive and IFN signal and as result can mount an innate defense mechanism [176,177]. Rotavirus can cause gastroenteritis and mainly infants are affected. For defense against Rotavirus all groups of interferons are important with IFNλ (type III IFN) being the most important one [175], since it is inducing antiviral defense of intestinal epithelial cells [178]. Furthermore, an interplay of ILC3-derived IL-22 and IFNλ has been described to limit mucosal virus infection [173].

Mouse Hepatitis Virus is a pathogenic virus, targeting the intestine and liver and causing severe systematic disease [175]. Gil-Cruz et al. showed that during MHV infection IL-15 released by fibroblastic reticular cells (FRCs) in a MyD88-dependent manner controls ILC1s and intestinal inflammation [179]. Specific knock-out of MyD88 in FRC led to hyperactivation of ILCs, which induce severe intestinal inflammatory disease characterized by commensal dysbiosis, loss of intestinal barrier function and decreased resistance to colonization. Therefore, in MNV infection FRC were shown to restrain activation of ILC1, thus preventing immunopathology [175]. This study again highlights the interplay of FRCs and ILCs, which are key to intestinal immune response not only against bacterial pathogens as described above, but also against viruses [179].

## 9. Conclusions

Due to research of the last decade, ILCs have emerged as crucial players in regulation of tissue homeostasis and innate immune responses. They are a heterogeneous family of innate immune cells and form the first line of defense, since they are strategically located at barrier surfaces, such as the intestinal mucosa. Due to their location, they can react rapidly to infectious pathogens by release of inflammatory mediators or interaction with other immune cells, before the adaptive immune response is activated. Despite their important role in infectious diseases, tissue homeostasis and tissue injury repair, ILCs can also contribute to chronic inflammation and tissue pathology. In an inflammatory state, distinct cytokines and tissue factors can change ILC composition. Depending on the signals from the microenvironment, ILCs can also transdifferentiate and change the outcome of the disease. This can be beneficial, since this plasticity allows for adaption to changing local conditions and thereby to a fine-tuned immune response to different pathogens. However, ILCs that do not transdifferentiate back may contribute to severe pathology during infection. Therefore, future work should focus on cytokines or other molecules that are beneficial for targeting ILCs in infectious diseases and possibilities to re-differentiate ILCs to their native state. These findings may then result in more specific targeting strategies for infectious disease treatments. 

## Figures and Tables

**Figure 1 ijms-23-14200-f001:**
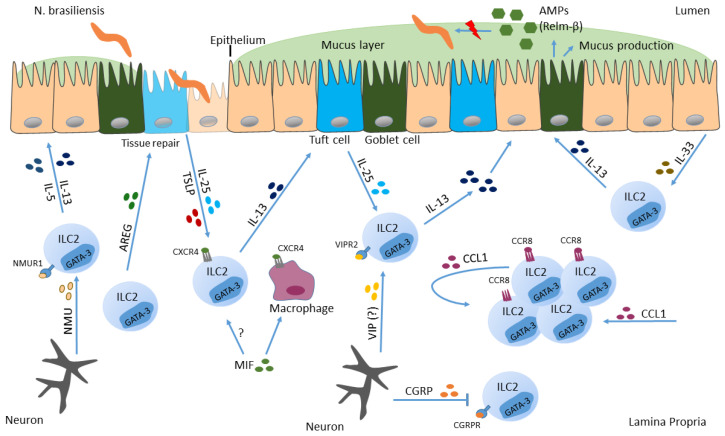
Action of ILC2s in *Nippostrongylus brasiliensis* infection. Helminths can infect barrier surfaces such as the small intestine. There, intestinal epithelial cells secrete IL-33, IL-25 and TSLP in response to pathogen invasion and cell destruction, which in turn activates ILC2s to secrete IL-13. In addition to epithelial cells, also neurons respond to NB and promote type 2 immune response. Furthermore, IL-13 induces goblet cells to secrete mucus, and induces tuft cell hyperplasia, resulting in expulsion of the parasite.

**Figure 2 ijms-23-14200-f002:**
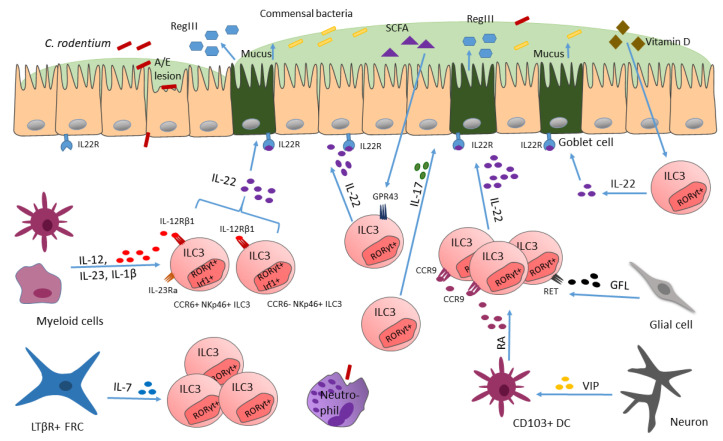
ILC3s reacting to *C. rodentium* invasion. Upon CR infection cytokines such as IL-23 and IL-1β are released, activating ILC3s to secrete IL-22. Thereafter, antimicrobial peptides (AMPs) are secreted (e.g., RegIII), which limits CR infection. RegIII secretion of intestinal epithelial cells is mediated via ILC3-derived IL-22. Transcription factors such as IRF-1 can control ILC3 numbers. Additionally, VIP secreted by neurons, induces CD103+ dendritic cell secretion of RA, required for ILC3 homing to the intestine. Furthermore, by release of IL-7, LTβR+ FRCs are required for sustenance of ILC3 numbers, which is important for mounting proper host defense against CR.

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
