# Peer review of "ILCs—Crucial Players in Enteric Infectious Diseases"

_ijms, 2022, doi:10.3390/ijms232214200_

Round 1
Reviewer 1 Report
The authors provide a very broad overview of ILCs. They present the types of ILCs in an understandable way, summarize the factors that regulate their plasticity, and discuss their role in intestinal homeostasis. Through each type of ILC, the ILC response to the infection caused by each GI pathogen and its consequences at the tissue level are presented in a clear and understandable way.
The literature used is relevant, and the figures are illustrative.
The use of English is appropriate; minor language correction is recommended only. The presentation of ILCs and the exploration of their diverse functions may form the basis for a number of future basic research projects and therapeutic target identification efforts.
The article is certainly recommended for acceptance.
Reviewer 2 Report
Overall the manuscript is a well written and in depth review of the participation ILC function and phenotype in gut infections of various pathogens. The paper is long and seems to be a reasonably complete review of what has been published on the topic. it would be useful for people who in particular are interested in ILC function or in the immune response to one of the gut pathogens in which the role of ILCs is reviewed. Line 33 is clumsy and needs a minor rewrite because the word "habitates" does not exist in the dictionary, likely the proper word would be "inhabits" but the sentence structure needs to be fixed as well.